# Trpv4-mediated apoptosis of Leydig cells induced by high temperature regulates sperm development and motility in *zebrafish*

Yasuhiro Yamamoto [1✉], Daisuke Hishikawa[2] & Fumihito Ono [1✉]

Exposure of testes to high-temperature environment results in defective spermatogenesis. *Zebrafish* exposed to high temperature exhibited apoptosis not only in germ cells but also in Leydig cells, as expected from studies using mice or salmon. However, the role of testicular somatic cells in spermatogenesis defects remains unclear. We found that in Leydig cells the *Trpv4* gene encoding the temperature sensitive ion channel was specifically upregulated in high temperature. High temperature also reduced hormone synthesis in Leydig cells and led to a prompt downregulation of sperm motility. In the *Trpv4* null mutant, neither Leydig cell-specific apoptosis nor decreased sperm motility was observed under high temperature. These results indicate that Leydig cell specific-apoptosis is induced via *Trpv4* by high temperature. Notably this *Trpv4*-dependent mechanism was specific to Leydig cells and did not operate in germ cells. Because sperm exposed to high temperature exhibited compromised genome stability, we propose that temperature sensing leading to apoptosis in Leydig cells evolved to actively suppress generation of offspring with unstable genome.

[1] Department of Physiology, Osaka Medical and Pharmaceutical University 2-7, Daigaku-machi, Takatsuki, Osaka 569-8686, Japan. [2] Department of Biochemistry and Molecular Biology, Nippon Medical School, 1-1-5 Sendagi, Bunkyo-Ku, Tokyo 113-8602, Japan. ✉email: yasuhiro.yamamoto@ompu.ac.jp; fumihito.ono@ompu.ac.jp

Reproduction in an environment suitable for genome stability is important for the preservation of species. This requirement is more imperative in poikilotherms animals, in which gametes are directly exposed to undesirable temperatures. As a result, they may have evolved a system that rapidly adapts to such changes. In fish, for example, temperature is generally considered to be an important factor in determining the exact timing of gamete maturation and spawning[1].

Inhibition of reproduction by high temperature is observed in a wide range of fish species[2]. High-temperature environment inhibits female ovulation and male spermatogenesis of Atlantic salmon (*Salmo salar*), which leads to arrest[3]. The suppression of ovulation by the high-temperature environment is due to decreased expression of *20β-hsd*, which in turn inhibits maturation-inducing steroid conversion. However, the mechanism of spermatogenesis arrest with regard to the sensing of temperature in the testis remains unclear.

Testes in mammals as well as in fish show clear sensitivities to high-temperature environments. Mammalian testes exposed to temperature >37 °C display abnormal spermatogenesis by way of germ cell apoptosis through various signaling pathways[4]. A proposed model posits that high temperature causes damage to meiotic prophase I and damaged pachytene spermatocytes are eliminated at the checkpoints that monitor meiotic progression[5]. Checkpoints in multicellular organisms induce apoptosis in DNA-damaged cells and contribute to genome stability across generations. This mechanism contributes to the lower mutation rate of germ cells than somatic cells, which is important for stronger genome integrity[6]. The mitotic checkpoint functions also in *zebrafish*: high temperature applied to knockouts of *Mps1*, which is required for cell cycle fidelity and genome stability leads to the failure of checkpoint and production of aneuploid sperm[7].

Leydig cells are found in the interstitial tissue of the testis and regulate spermatogenesis and reproductive function through the secretion of steroid hormones such as testosterone[8]. In mammalian models, hyperthermic stimulation of the testis by an artificially induced cryptorchidism leads to a decrease in Leydig cell activity as well as in germ cells[9]. Reduced activity of Leydig cells leads to decreased steroid hormone secretion[10,11]. High temperature also induces apoptosis not only in germ cells but also in Leydig cells[12]. However, the molecular mechanism and the significance of apoptosis in Leydig cells induced by high-temperature stimulation remain unexplored. In particular, the causal relationship between reduced Leydig cell activity and abnormal spermatogenesis due to the high-temperature environment is not clear.

In this study, we provide evidence that Leydig cells in *zebrafish* undergo apoptosis prior to germ cells when placed in high temperatures, and *Trpv4* is involved in this process. Furthermore, we show that the Leydig cell-specific apoptosis decreased the synthesis of steroid hormones, which in turn impaired the motility of sperm whose genome integrity is compromised.

## Results

**High temperature affects Leydig cells prior to germ cells**. To determine the appropriate condition for high-temperature stimulation in adult *zebrafish* whose optimal temperature is 28–30 °C, we first examined fish survival during 2 weeks of incubation at 33–39 °C. When placed in water <34 °C, mortality was not observed. In contrast, temperature >35 °C caused the death of incubated fish. Based on this observation, 34 °C was chosen as the temperature of stimulus in subsequent analyses (Supplementary Fig. 1). No abnormalities in spermatogenesis or morphological changes in the testes were observed after 2 weeks of rearing at 29 °C in the identical setup (Supplementary Fig. 2).

To analyze the effect of temperature stimulation on testicular structures and spermatogenesis, sections of the testes were observed by HE staining (Fig. 1a, Supplementary Fig. 3a). There are three developmental stages of spermatids: initial (E1), intermediate (E2), and final (E3)[13]. Temperature stimulation led to a decrease in spermatids, especially at E3 (circled in black lines), and the appearance of abnormal cells (circled in red lines). The abnormal cells had large nuclei relative to the cytoplasm and were different in appearance from any normal types of cells in germ cell development (Supplementary Fig. 4). Germ cells at pachytene, diplotene, and metaphase stages were present, but the number of spermatids was dramatically reduced in testis exposed to 34 °C. In normal *zebrafish*, germ cells at the identical differentiation stage are confined in a cyst[14]. In contrast, abnormal cells were observed within a single cyst containing meiotic germ cell at metaphase or spermatid after 1 and 3 days of exposure to 34 °C (Supplementary Fig. 4). After 7 days or 2 weeks at 34 °C, cysts with a mixture of normal and abnormal cells disappeared and only those with abnormal cells were observed (Supplementary Fig. 4). These results indicate that under high-temperature differentiation from metaphase to spermatid stages became abnormal.

In addition, there was a significant decrease in the interstitial tissue, which was observed as gaps between cysts. This tendency became more obvious as the duration of exposure to 34 °C became longer. Specifically, distinct Leydig cells were rarely detected after 2 weeks of exposure to 34 °C (Fig. 1a, upper panels).

Transfer of fish from 34 °C back to 29 °C resulted in the disappearance of the abnormal cell population and the appearance of interstitial cells in 7 days (Fig. 1a, lower panels). E3 spermatid appeared within 10 days of the transition to 29 °C, and spermatogenesis completely recovered to normal in 1 month. In *zebrafish* testes, undifferentiated type-A spermatogonia and differentiated type-B spermatogonia are present[14]. The type-A spermatogonia differentiates from E3 spermatid in 7 days[13]. There were no morphological abnormalities observed in type-A and type-B spermatogonia in testes treated at 34 °C temperature for 2 weeks (Supplementary Fig. 3b). Therefore, we expected that the transition to 29 °C would induce differentiation to the E3 spermatid within 1 week. However, differentiation into E3 spermatid was observed only after 10 days of recovery, which suggests the involvement of factors other than autonomous differentiation of germ cells.

We analyzed the expression of several groups of genes in testes exposed to 34 °C. Among them, genes in the *Heat shock protein* (*Hsp*) group showed significantly upregulated expression within 24 h of temperature stimulation (Fig. 1b). *Hsps* are genes that function as molecular chaperones and protect cells when exposed to stress conditions such as heat[15]. Most of the Leydig cell marker genes were significantly downregulated in the testes at 3 days of temperature stimulation. In contrast, the majority of marker genes for germ cells and Sertoli cells showed no change (Fig. 1b).

Based on the loss of E3 spermatid at 34 °C (Fig. 1a), we predicted a decrease in *Odf3b*, a spermatid marker[16], which contrary to our expectation was not changed (Fig. 1b). To investigate why *Odf3b* expression was not decreased at 34 °C, we analyzed the localization of *Odf3b* expression in testes. *Odf3b* was expressed in spermatocytes and early spermatids but not in E3 spermatid at 29 °C (Supplementary Fig. 5). In contrast, it was expressed in spermatocytes and abnormal cells at 34 °C (Supplementary Fig. 5). These results suggested that temperature stimulation led to differentiation of Odf3b-expressing abnormal spermatid, which was consistent with histology (Fig. 1a).

Since temperature stimulation strongly affected Leydig cells, they were further analyzed by in situ hybridization using its

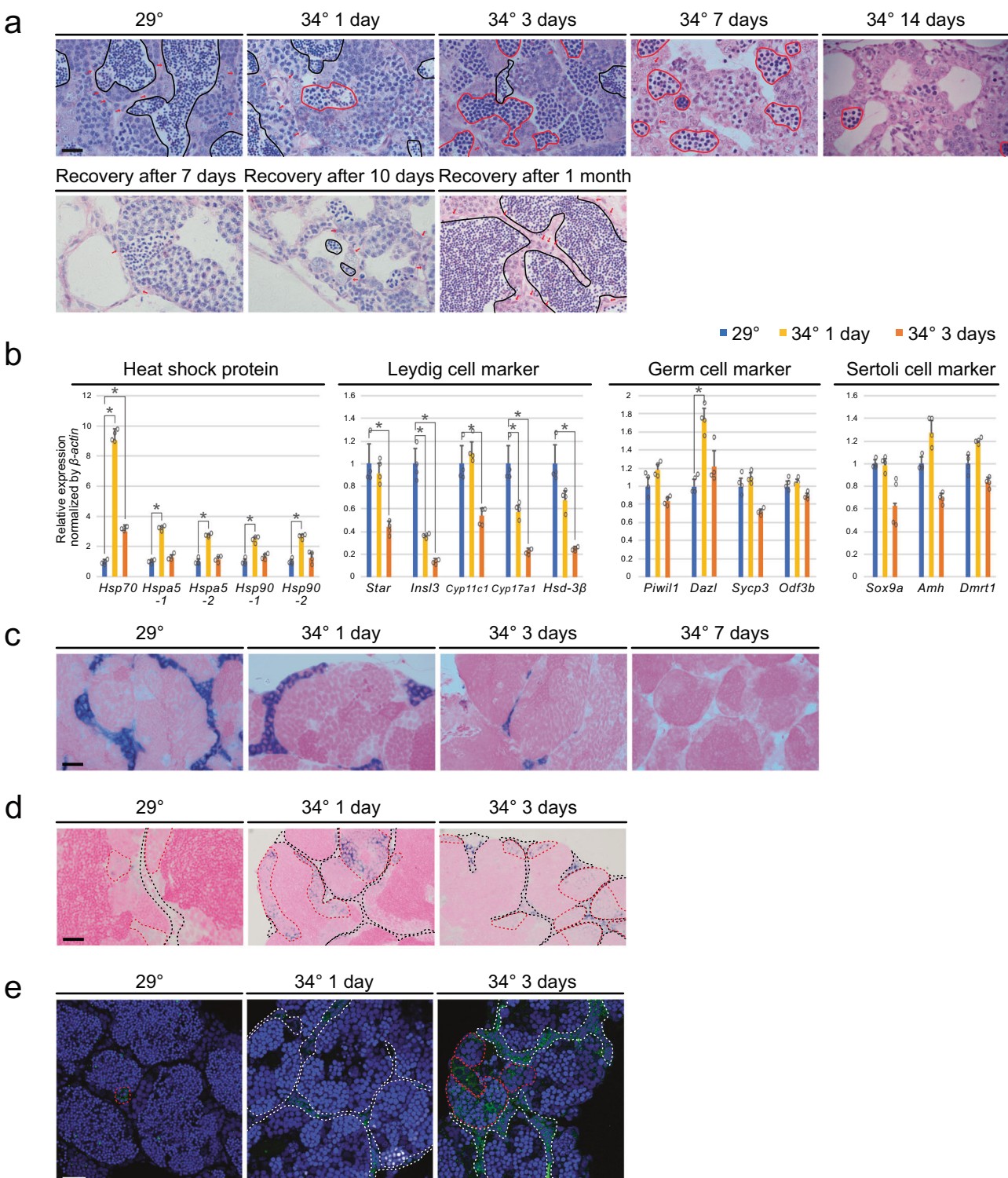

**Fig. 1 High-temperature treatment led to significant apoptosis of Leydig cells. a** Top row: HE staining of testes. The black lines indicate areas containing E3 spermatid. The red lines indicate areas containing abnormal cell populations. Arrows indicate interstitial tissue. Interstitial areas decreased at 3, 7, and 14 days of temperature stimulation. Bottom row: Recovery of spermatogenesis and interstitial areas from temperature stimulation. The abnormal cell population disappeared, and E3 spermatid and interstitial areas were observed. **b** qRT-PCR analysis in testes. Values are the mean ± SEM ($n = 4$). *$p < 0.01$ compared with the 29 °C testes. **c** In situ hybridization of *Insl3*, a Leydig cell marker, in testes. **d** In situ hybridization of *Hspa5*. Black broken lines indicate interstitial cells and red broken lines indicate spermatocyte cells. **e** Localization of cleaved caspase 3. White broken lines indicate interstitial cells and red broken lines indicate spermatocyte cells. Scale bars are 20 µm.

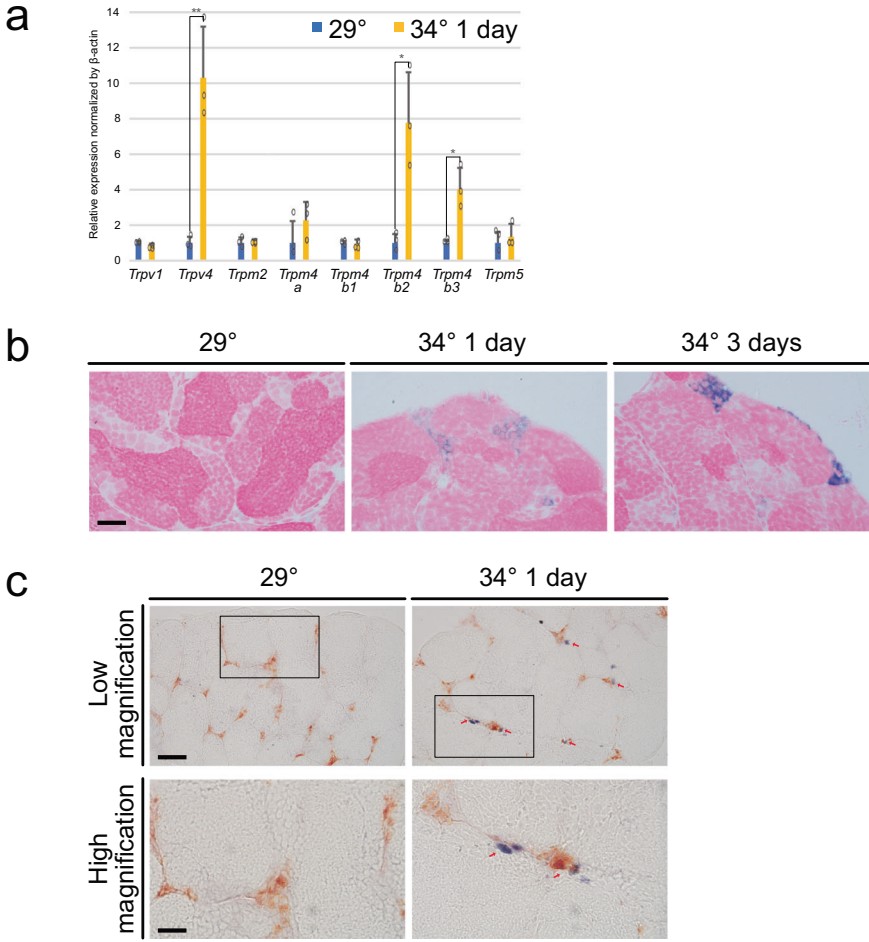

**Fig. 2 Higher temperature led to upregulated expression of *Trpv4* in Leydig cells. a** qRT-PCR analysis of Trp family genes in the testes. Values are the mean ± SEM ($n = 3$). **$p < 0.01$ and *$p < 0.05$ compared with the 29 °C testis. **b** In situ hybridization of *Trpv4* in testes. **c** Double in situ hybridization of *Trpv4* (blue) and *Insl3* (yellow). The insets were magnified in the bottom rows. Scale bars are 20 μm (**b** and **c**: high magnification) and 50 μm (**c**: low magnification).

marker *Insl3*[17]. The region with positive signal decreased as the duration of 34 °C treatment became longer (Fig. 1c), which was consistent with HE staining and the expression analysis (Fig. 1a and b). Of note, the expression of *Hspa5*, involved in protein folding and assembly in the endoplasmic reticulum (ER)[18] was induced by high temperature not only in Leydig cells but also in germ cells (Fig. 1d). These results suggest that the ER stress response to temperature stimulation occurred in both Leydig cells and germ cells.

Suspecting that apoptosis is involved in the different responses between Leydig cells and germ cells, we analyzed the localization of its marker, cleaved caspase 3. The signal was detected in the interstitial cells (circumscribed in white lines) after 1-day exposure to 34 °C and in spermatocytes (in red lines) after 3 days (Fig. 1e). These results indicated that apoptosis in response to 34 °C exposure is induced in Leydig cells prior to spermatocytes.

**Temperature stimulation led to increased expression of *Trpv4* in Leydig cells**. To identify molecules involved in temperature sensing, we performed qRT-PCR for a group of *Trp* channel genes, sensor molecules for thermoreception[19]. Transient receptor potential (TRP) channels are expressed in sensory organs and play an important role in temperature sensitivity in *zebrafish*[20,21]. Expression of *Trpv4*, *Trpm4b2*, and *Trpm4b3* was significantly increased by temperature stimulation (Fig. 2a). Among the three, the change was most evident for *Trpv4*, which was upregulated

specifically in interstitial cells as shown by in situ hybridization (Fig. 2b). To further characterize these *Trpv4*-expressing cells, we performed double in situ hybridization using Leydig cell marker genes *Insl3* and *Trpv4*. Remarkably, *Trpv4* expression (blue) was limited to *Insl3*-positive cells (brown) in testes treated at 34 °C (Fig. 2c). These results show that temperature stimulation led to increased expression of *Trpv4* specifically in Leydig cells.

**In *Trpv4* KOs, apoptosis of Leydig cells was not induced by high-temperature**. To clarify the function of TRPV4, a *Trpv4* mutant was generated using the CRISPR-Cas9 system (Supplementary Fig. 6). An 8 bp deletion in exon 2 resulted in a frameshift causing a change in amino acid sequence and a premature stop codon. *Trpv4*$^{-/-}$ zebrafish were fertile and no abnormalities in spermatogenesis were observed under normal rearing conditions.

To analyze the effects of high-temperature treatment on testis structure and spermatogenesis in *Trpv4*$^{-/-}$, testis sections were observed with HE staining (Fig. 3a, Supplementary Fig. 7). After 2 weeks at 34 °C, *Trpv4*$^{-/-}$ as well as *Trpv4*$^{+/+}$ showed a decrease in E3 spermatid (black lines) and the appearance of abnormal cells (red lines). In contrast, distinct effects were observed in stromal tissue: *Trpv4*$^{+/+}$ showed reduced interstitial tissue (arrows in Fig. 3a), while in *Trpv4*$^{-/-}$ it remained unaffected. Moreover, recovery of E3 spermatid was observed as early as

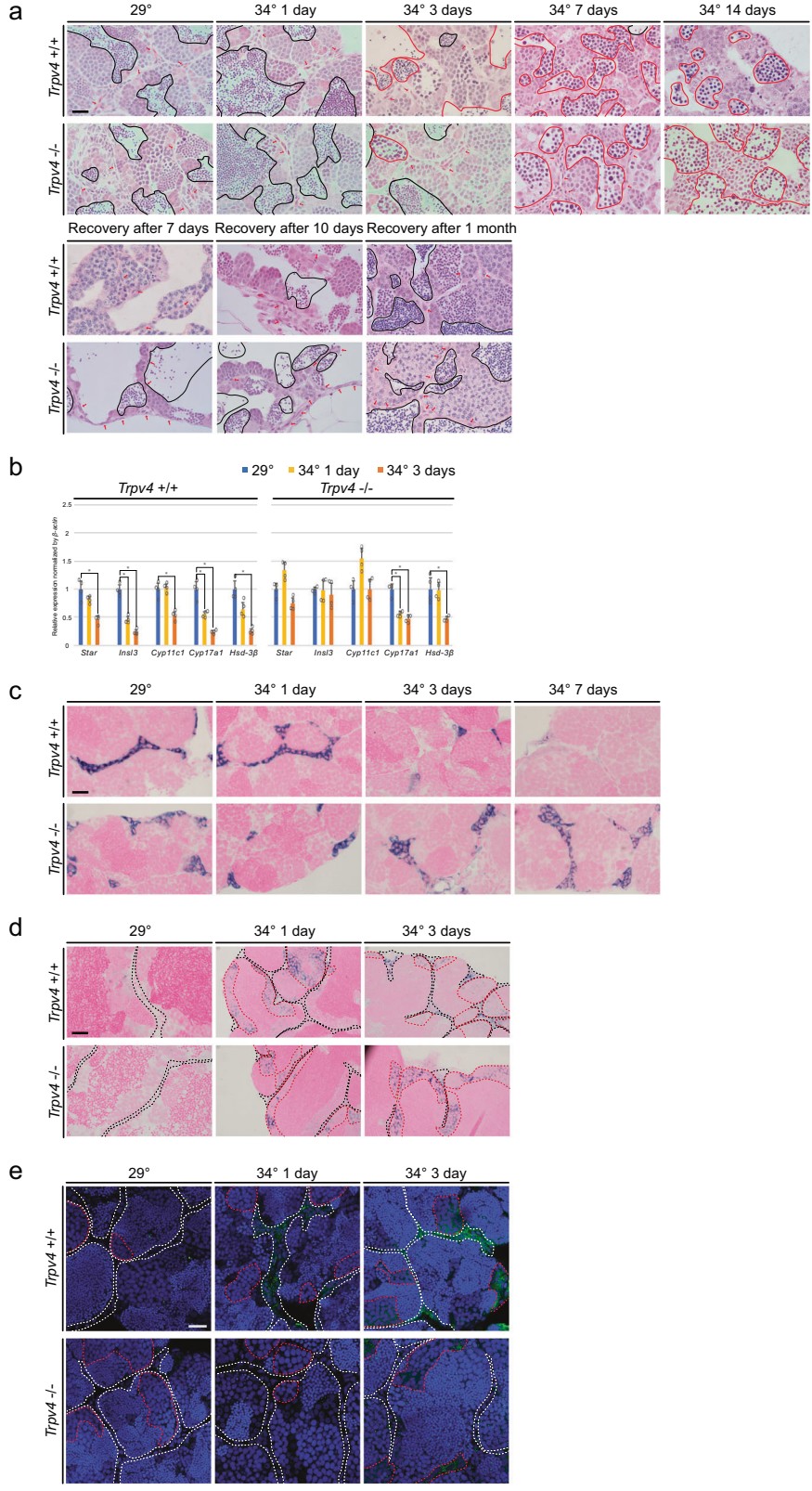

**Fig. 3 Leydig cells in *Trpv4* KOs did not undergo apoptosis in high temperature. a** HE staining of testes. The upper panels show *Trpv4*[+/+] and the lower panels show *Trpv4*[−/−]. Black line indicates areas of E3 spermatid. Red line indicates an abnormal cell population. Arrows indicate interstitial tissue. Interstitial tissue can be detected clearly in *Trpv4*[−/−] after 7 and 14 days of temperature stimulation. **b** qRT-PCR analysis of Leydig cell marker genes in testes. Values are the mean ± SEM (*n* = 4). *p < 0.01 compared with the 29 °C testis. **c** In situ hybridization in testes of *Isl3*, which is a Leydig cell marker. **d** In situ hybridization of *Hspa5*. Black broken lines indicate interstitial cells and red broken lines indicate spermatocyte cells. **e** Localization of cleaved caspase 3. White broken lines indicate interstitial cells and red broken lines indicate spermatocyte cells. Scale bars are 20 μm.

7 days after transferring the fish from 34 to 29 °C, earlier in $Trpv4^{-/-}$ than in $Trpv4^{+/+}$.

To examine further the effects of $Trpv4$ KO on Leydig cells, the expression of the Leydig cell marker genes was examined. *Star*, *Insl3*, and *Cyp11c1* were significantly decreased by treatment at 34 °C in $Trpv4^{+/+}$, while they remained unchanged in $Trpv4^{-/-}$ (Fig. 3b). Expression of *Cyp17a1* and *Hsd3β* was significantly decreased in $Trpv4^{-/-}$ as well as $Trpv4^{+/+}$ at 34 °C. These results suggested that steroid hormone biosynthesis in $Trpv4^{-/-}$ was affected to a lesser extent by temperature stimulation. Heterogeneity of Leydig cells in *zebrafish*, as was reported in adult mice[22], may have affected the expression of marker genes in response to high-temperature stimulation. Noticeable difference was not observed for *Hsp*, germ cell markers or Sertoli cell markers (Supplementary Fig. 8). Correspondingly, *Insl3*(+) area decreased as the duration of treatment at 34 °C increased in $Trpv4^{+/+}$, while it remained unchanged in $Trpv4^{-/-}$ (Fig. 3c). Notably, the expression of *Hspa5* showed no difference (Fig. 3d). Strong signals of cleaved caspase3 were detected in the interstitial cells (white lines) and spermatocyte (red lines) after 3 days of 34 °C treatment in $Trpv4^{+/+}$. In $Trpv4^{-/-}$, in contrast, cleaved caspase3 was restricted to the spermatocytes, and no clear signal in the interstitial cells was observed (Fig. 3e). These results indicate that temperature stimulation failed to induce Leydig cell apoptosis in $Trpv4^{-/-}$.

**High temperature impaired sperm motility via *Trpv4* in Leydig cell**. We analyzed the involvement of *Trpv4* in the motility of mature spermatozoa by exposing adult males to 34 °C for 1 day. When the motility of sperm was analyzed under SI8000, which allows quantification of cell movement, a significant decrease in swimming velocity was observed in $Trpv4^{+/+}$ but not in $Trpv4^{-/-}$ (Fig. 4a, c and Supplementary Movies 1–4, short movies showing representative cases for each genotype). The analysis of trajectories revealed that sperm from $Trpv4^{+/+}$ after 34 °C exposure followed small circular trajectories, traveling shorter distance, which was not observed in $Trpv4^{-/-}$ (Fig. 4b). Steroid hormone (17α,20β,21-trihydroxy-4-pregnen-3-one (20β-S)) secreted from fish testis work on receptors expressed in sperms and control their motility[23–26]. 20β-S at 100 nM resulted in increased motility and improved trajectory of spermatozoa from $Trpv4^{+/+}$ after 34 °C exposure (Fig. 4d–f and Supplementary Fig. 9).

Based on these results, we hypothesized that the decrease of 20β-S secreted from Leydig cells after 34 °C treatment impaired sperm motility. The secretion of 20β-S in the testis was analyzed by LC–MS/MS. Unfortunately, 20β-S in the testes was below the detection limit (data not shown). We therefore used qRT-PCR to examine the expression of the *20β-hsd* by qRT-PCR, an enzyme involved in the synthesis of 20β-S[27]. In $Trpv4^{+/+}$, the expression was significantly decreased in testes at 3 days of temperature stimulation, while no difference was observed in $Trpv4^{-/-}$ (Fig. 4g). The result was further confirmed by in situ hybridization. In $Trpv4^{+/+}$, *20β-hsd* was expressed in the interstitial cells at 29 °C, but its expression disappeared as the incubation at 34 °C lasted longer. In contrast, $Trpv4^{-/-}$ showed clear expression in the interstitial cells even after temperature stimulation (Fig. 4h). Based on these results, we propose that the reduction of 20β-S synthesis, caused by decreased expression of *20β-hsd* in Leydig cells, is responsible for the decrease in sperm motility induced by high temperature.

Indeed, when sperm isolated from the testis was directly exposed to high temperature, its motility did not decrease even at 40 °C (Supplementary Fig. 10). Although the duration of exposure to high temperature was shorter than the incubation of the whole

fish (Fig. 4) due to technical limitations of maintaining isolated sperm in vitro, these results were consistent with the hypothesis that the endocrine regulation of Leydig cells, not autonomous regulation of sperm, determined the sperm motility in response to high temperature.

**Sperm matured in a high-temperature environment leads to abnormalities in offspring**. We hypothesized that *zebrafish* actively inhibit sperm motility in high temperatures so that embryos containing damaged cells, in particular damaged gametes, will not be generated. To test these predictions, we used flow cytometry to analyzed the chromosome content of sperm after 1 day of exposure to 34 °C. Regions of small cell size and low PI content were defined as mature sperm fractions (Supplementary Fig. 11). Sperm after 1-day exposure to 34 °C displayed a wider range of DNA content than sperm at 29 °C (Fig. 5a and b).

Males incubated at 34 °C for 1 day were mated with females normally reared and incubated at 29 °C. Fertilization rates were significantly lower for both WT and $Trpv4^{-/-}$ incubated at 34 °C than at 29 °C. However, in the 34 °C group, WT had significantly lower fertilization rates than $Trpv4^{-/-}$ (Fig. 5c). We propose that the $Trpv4^{-/-}$, whose sperm velocity did not decrease even at 34 °C, had a decrease in fertilization rate smaller than that of WT (Supplementary Fig. 12). Among the fertilized eggs, there was no difference in survival up to 6 days post fertilization (dpf) (Fig. 5d). However, higher developmental abnormalities were observed in offspring from $Trpv4^{+/+}$ and $Trpv4^{-/-}$ males exposed to 34 °C (Fig. 5e, f). Because a high frequency of developmental abnormalities occurs in offspring derived from aneuploid sperm in *zebrafish*[7,28], these results suggested that some of the spermatozoa that matured in high-temperature environments have compromised genome stability. They also suggested that *Trpv4* is not involved in the maintenance of sperm quality.

If fertilization occurs at high temperatures, generated embryos are also likely to develop in high temperatures. We therefore examined the effect of 34 °C water temperature on developing embryos of *zebrafish*. Embryos incubated at 34 °C resulted in high mortality rates and high developmental anomalies exceeding 70% at 1 dpf (Supplementary Fig. 13). It was clear that 34 °C is not a favorable temperature for normal embryonic development. However, it is of note that some embryos do develop normally and potentially grow up into fertile adults. These results indicated that *zebrafish* actively reduce sperm motility in order to suppress fertilization in high temperatures and thereby prevent the creation of embryos with compromised genome stability that can potentially be transmitted to offspring.

**Temperature-dependent regulation of Leydig cells through *Trp* is species-specific**. Finally, to examine whether this mechanism of Leydig cell sensitivity to high temperature is universally conserved among fish, we analyzed *medaka* (*Oryzias latipes*), which had an optimal temperature range distinct from that of *zebrafish*[29,30].

Similar to *zebrafish*, high-temperature stimulation in *medaka* testes leads to abnormal spermatogenesis[31]. However, the optimal temperature for spermatogenesis was also different between the two species. *Medaka* required exposure to higher temperatures compared to *zebrafish* to show reduced sperm motility (Supplementary Fig. 14a and b).

We performed qRT-PCR analysis of a group of *Trp* channels in the testes of *medaka* exposed to 39 °C. *Hsp* group was significantly upregulated in the temperature stimulation, but none of the genes in the *Trp* group was significantly upregulated (Supplementary Fig. 14c). Cleaved caspase3 was widely observed in the Leydig cells and germ cells especially spermatocytes after

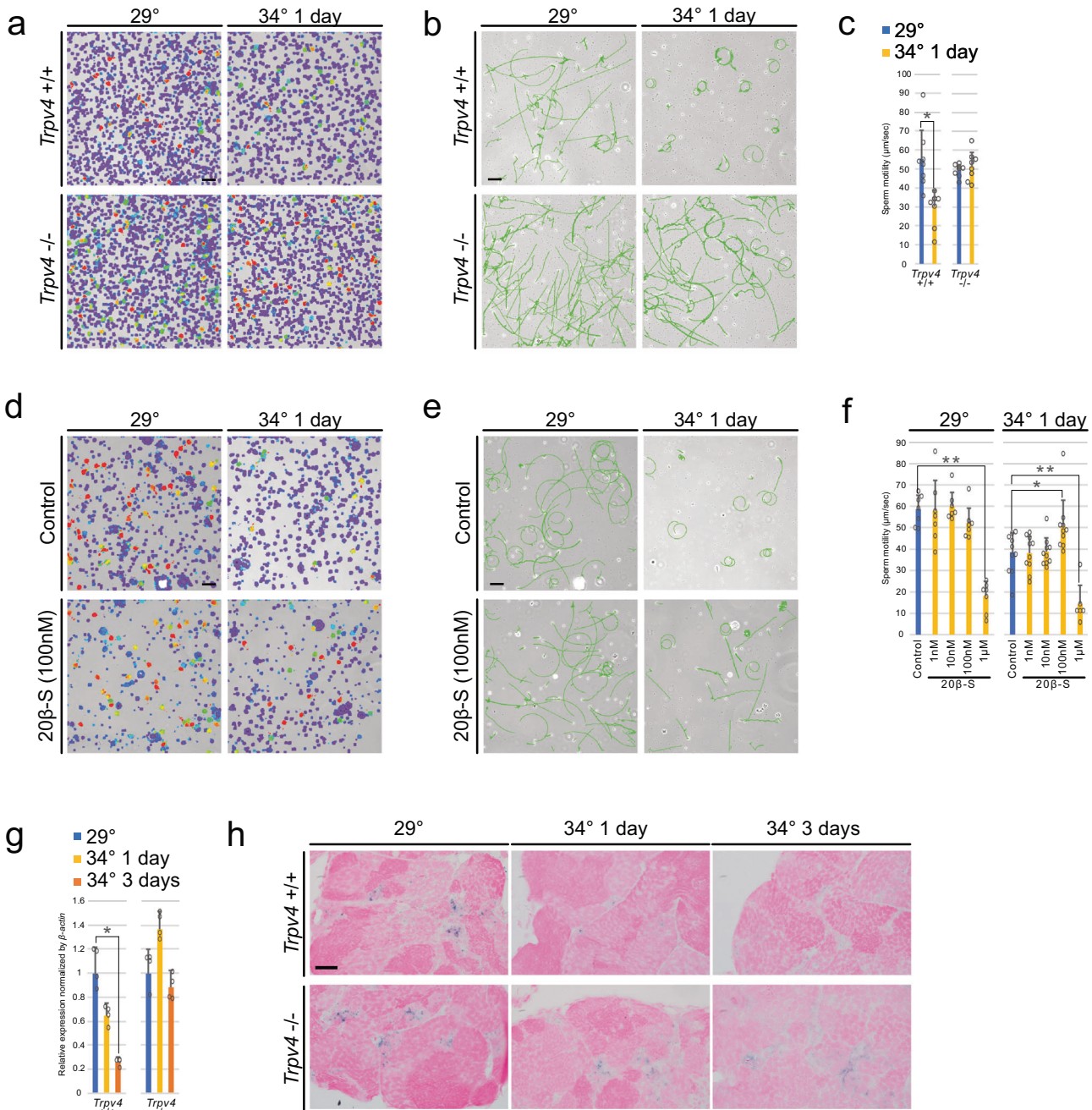

**Fig. 4 Higher temperature impaired sperm motility in *Trpv4*[+/+] but not in *Trpv4*[−/−]. a** Heat map images of sperm motility. Red indicates high motility. Light blue indicates low motility. **b** Tracking of sperm swimming. The green line shows the trajectory of sperm in 3 s. **c** Pooled data of sperm motility in two groups. Values are the mean ± SEM ($n = 8$). *$p < 0.01$ compared with the 29 °C. **d** Heat map images of sperm motility with the addition of 100 nM 20β-S. Red indicates high motility; light blue indicates low motility. **e** Tracking of sperm swimming with the addition of 100 nM 20β-S. **f** Averaged sperm motility in added 20β-S. Values are the mean ± SEM (29 °C $n = 7$ and 34 °C $n = 6$). **$p < 0.01$ and *$p < 0.05$ compared with the 29 °C. **g** qRT-PCR analysis of the *20β-Hsd* gene in the testes. Values are the mean ± SEM ($n = 4$). *$p < 0.01$ compared with the 29 °C testis. **h** In situ hybridization of *20β-hsd*. Scale bars are 50 μm (**a**, **b**, **d**, and **e**) and 20 μm (**h**).

1 day of exposure to 39 °C (Supplementary Fig. 14d). These results indicate that the induction of acute apoptosis in Leydig cells via *Trpv4* by temperature stimulation is a phenomenon characteristic to *zebrafish*, and *medaka* may employ different molecules to reduce sperm motility in high temperature[31].

## Discussion
Multiple reports in mammals showed that germ cells are the main target of detrimental high temperatures in testes[32]. Our study using *zebrafish* revealed that high temperature induced apoptosis

in Leydig cells prior to germ cells. Moreover, we showed that Leydig cells and germ cells in *zebrafish* have distinct mechanisms in sensing and responding to high temperatures. *Trpv4* was involved in the Leydig cell-specific high-temperature sensitivity, which caused apoptosis, leading in turn to impaired sperm motility and reduced fertilization.

TRP channels were identified as sensor molecules for thermal reception, through which cellular responses to temperature changes are regulated[33]. Each TRP channel has a unique thermoreceptive range. Although TRP proteins are expressed also

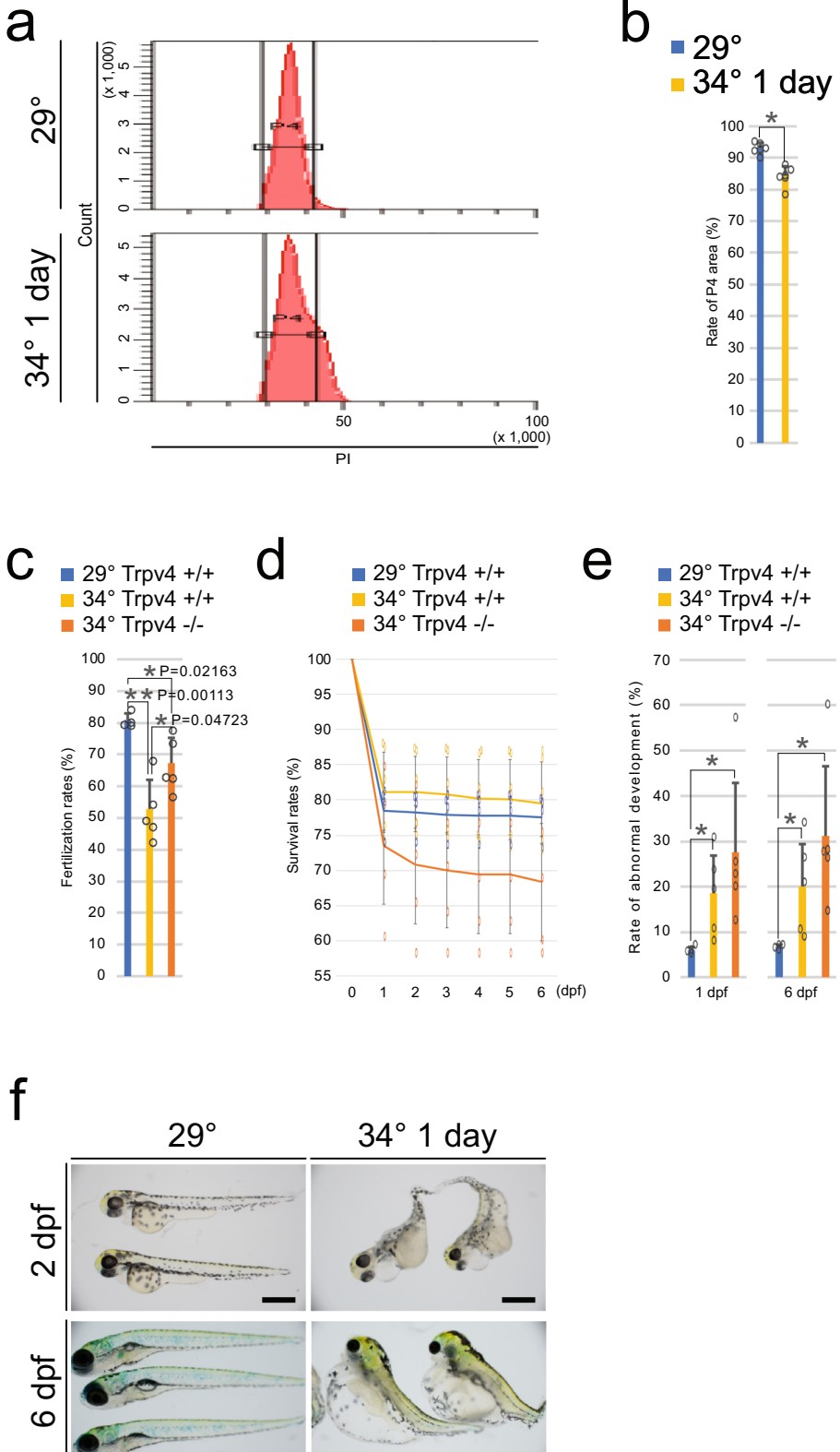

**Fig. 5 Sperm matured at high temperature showed an abnormality. a** Flow cytometric analysis of sperm DNA content. PI luminance values (30–42) were set as the P4 gate. **b** Percentage of sperm within the P4 gate. Values are the mean ± SEM ($n = 6$). *$p < 0.01$ compared with the 29 °C sperm. **c** Fertilization rates of 34 °C exposed male. Values are the mean ± SEM (29 °C $n = 4$ and 34 °C $n = 5$). **$p < 0.01$ and *$p < 0.05$ compared two groups. **d** Survival rates of embryos derived from 34 °C exposed males. Values are the mean ± SEM (29 °C $n = 4$ and 34 °C $n = 5$). *$p < 0.01$ compared with the 29 °C sperm.
**e** Percentage of morphologically abnormal embryos derived from 34 °C exposed males. Values are the mean ± SEM (29 °C $n = 4$ and 34 °C $n = 5$).
*$p < 0.01$ compared with the 29 °C sperm. **f** Images of embryos at 2 dpf and 6 dpf. Scale bars are 500 µm.

in the testes[34], *Trp* in the *zebrafish* testis is poorly characterized. *Trpv4* is strongly expressed in stressful environments[35,36] and activated above 27–35 °C[37], which fits the high-temperature condition used in this study. *Trpv4* activation by endogenous and exogenous stimuli increases $Ca^{2+}$ influx, resulting in an excess of intracellular free $Ca^{2+}$[38], and causes apoptosis in pathological conditions[35,36,39]. Trpv4 in *zebrafish* was studied by Amato et al. who showed the expression of this protein in sensory organs[21]. The lack of temperature sensing mechanism may also contribute to the phenotypes of *Trpv4^{-/-}* we observed (Fig. 3). Taken together, these characteristics of *Trpv4* fit our hypothesis that high temperature led to upregulation of *Trpv4* in Leydig cells and caused apoptosis.

In *zebrafish*, as in mice, high-temperature stimulation caused abnormal spermatogenesis (Fig. 1a). Specifically, high-temperature treatment induced apoptosis in spermatocytes and abnormal differentiation from spermatocyte to spermatid, which was consistent with mice. Meiotic checkpoints cause spermatogenesis defects in mice following high-temperature stimulus and eliminate damaged spermatocytes[40]. A similar, sperm cell-autonomous mechanism seems to operate also in *zebrafish*. Hormone secretion from Leydig cells is unlikely to be essential for these cell-autonomous apoptoses because spermatogenesis failure was not distinguishable between *Trpv4^{+/+}* and *Trpv4^{-/-}* (Fig. 3, Supplementary Fig. 12). Indeed, knockouts of steroid synthase genes in *zebrafish* can produce mature spermatozoa[41–44], which was consistent with our hypothesis. On the other hand, the effect of hormones secreted from Leydig cells on certain aspects of spermatogenesis was suggested from our data. First, the transition from 34 to 29 °C led to earlier E3 spermatid differentiation in *Trpv4^{-/-}* compared to *Trpv4^{+/+}*. Second, expression of several genes including *Insl3* did not change in *Trpv4^{-/-}* exposed to 34 °C. In *zebrafish*, *Insl3* was reported to act as a germline survival factor[42]. In *Trpv4^{-/-}*, hormone secretion from Leydig cells was likely involved in promoting the rapid differentiation of E3 spermatid or survival of the remaining germ cells.

We propose that the Leydig cell-specific temperature-sensing mechanism comprises a system that suppresses fertilization in *zebrafish* under high-temperature conditions. In their natural habitat, *zebrafish* live in shallow tropical water with slow currents that are subject to frequent temperature fluctuations[45], where water temperatures can rise to 38.6 °C[46]. High-temperature environments lead to severe abnormalities in early *zebrafish* development (Supplementary Fig. 13). Importantly, the maturation of spermatozoa in a high-temperature environment resulted in aneuploid spermatozoa (Fig. 5a). In addition, a high-frequency of developmental abnormalities were observed in offspring, even when incubated at 29 °C after fertilization (Fig. 5e and f). If gametes with compromised genomic integrity are transmitted to the next generation, the result will be detrimental to the species. Therefore, a mechanism to avoid fertilization in high-temperature environments will be advantageous.

Because the meiosis checkpoint controls the differentiation from metaphase to early spermatid, spermatozoa that have matured prior to temperature stimulation cannot be eliminated by the meiosis checkpoint. (Supplementary Figs. 4 and 5). Indeed, E3 spermatids were abundantly observed in the testis after 1-day incubation at 34 °C (Fig. 1a and Supplementary Fig. 3). Moreover, mature spermatozoa did not lose sperm motility in high temperatures (Supplementary Fig. 10). Therefore, mature sperm differentiated prior to temperature stimulation are expected to be motile with fertilization capabilities even at 34 °C. However, sperm motility and fertilization rate were significantly reduced by a 1-day exposure to 34 °C (Figs. 4a, c and 5c). Taken together, we propose that *Trpv4*-mediated reduction of 20β-S synthesis is the mechanism responsible for inhibiting fertilization of mature

sperm which has already passed the meiosis checkpoint prior to exposure to high temperature.

Neither upregulated expression of *Trp* family genes nor Leydig cell-specific apoptosis at high temperatures was observed in *medaka*. Temperature stimulation at 39 °C resulted in reduced sperm motility and induction of apoptosis, especially in spermatocytes (Supplementary Fig. 14), while molecular changes identified in *zebrafish* were not observed in *medaka*. These results suggest that factors other than *Trp* and meiotic checkpoints are involved in the apoptotic response to high temperature in *medaka*. We propose that response to high temperature in Leydig cells was established as a system to actively suppress fertilization in *zebrafish*, whose adaptive water temperature range matched the molecular characteristics of *Trpv4*.

## Methods

**Fish**. All experiments using animals in this study followed the guidelines of Osaka Medical and Pharmaceutical University. Adult *zebrafish* and *medaka* were maintained at 28.5 °C on a 10 h light:14 h darkness photoperiod. *Zebrafish* were bred from a RIKEN WT background and *medaka* strain was OKcab. Male adults 6–12 months old were used for experiments.

**Temperature stimulation**. Up to four individuals were temperature-stimulated in W22 × D12 × H11 cm tanks with 1.8 l of water, and water changes were performed daily. The tanks were placed in a water bath at 35 °C and oxygen was supplied by aquarium air stones. Temperature stimulation was interrupted for feeding from 12:00 to 13:00 daily. Exposure of isolated sperm was for 20 min (Supplementary Fig. 9).

**Histology**. Testes were fixed in Bouin solution, and 4 μm plastic sections were prepared using Technobit 8100 (Heraeus Kulzer). Samples were stained with hematoxylin solution for 10 min (Muto pure chemicals, Tokyo, Japan), and stained with 1% eosin (Muto pure chemicals, Tokyo, Japan) for 30 s. The mounted samples were imaged using an Olympus DP74 camera (Evident, Tokyo, Japan).

**qRT-PCR**. RNA samples were extracted from testes. According to the manufacturer's instructions, 0.5 μg RNA template was used for reverse transcription to synthesize cDNA using a first-strand cDNA synthesis kit (ReverTra Ace, TOYOBO, Tokyo, Japan). The qPCR primers are listed in Supplementary Table 1. For amplification, the KOD SYBR qPCR/RT (TOYOBO, Tokyo, Japan) and ABI real-time system were used. Statistical significance was evaluated by two-tailed Student's *t*-test.

**In situ hybridization**. RNA in situ hybridization (WISH) was carried out following a standard protocol[47]. The primers used to amplify the regions used for the probes are shown in Supplementary Table 1. Briefly, testes were fixed in 4% paraformaldehyde in 1× PBS overnight. On the following day, samples were hybridized at 65 °C overnight. Samples were further incubated with anti-digoxigenin-AP antibody solution (1:2000) overnight at 4 °C and stained with NBT/BCIP. The stained samples were imaged using an Olympus DP74 camera (Evident). Double in situ hybridization was carried out following a standard protocol[48]. The first probe was labeled with DIG and the second probe was labeled with FITC. After the first staining, incubation was performed with 0.1 M glycine–HCl pH 2.2 for 40 min to remove alkaline phosphatase activity.

**Immunohistochemistry**. Immunofluorescence staining was performed using an anti-Cleaved caspase 3 antibody (#9661, Cell

Signaling Technology, USA). Alexa 488 was used as the secondary antibody. Nuclear DNA was stained with 4',6-diamidino-2-phenylindol (DAPI). Stained samples were imaged using Leica SP8 (Leica, Germany).

**Targeted genetic disruption of *Trpv4* by CRISPR/Cas9.** CRISPR/Cas9 target sites in *Trpv4* were searched using CRISPR/Cas9 target online predictor (crispr.direct)[49]. Selected target sites are shown in Supplementary Fig. 5. Guide RNA was generated using a Guide-it sgRNA In Vitro Transcription kit (TAKARA, Tokyo, Japan). Cas9 mRNA (50 ng/μl) and sgRNA (25 ng/μl) were injected into one-cell stage embryos. Mutant allele was identified in F1 adult fish using the primer sets shown in Supplementary Table 1 and sequence analysis.

**Sperm motility measurement.** *Zebrafish* and *medaka* testes were diluted in fetal bovine serum (FBS). In *zebrafish*, sperm diluted in FBS were activated by adding an equal volume of breeding water immediately prior to measurement. Sperm motility was recorded as sequential 1024 × 1024-pixel phase-contrast images with a ×10 objective lens for 3 s at 300 frames per second (fps) using cell motion system SI8000 (Sony Corporation, Tokyo, Japan). Images were acquired with SI8000 view Software (Sony Corporation, Tokyo, Japan) and analyzed with SI8000R Analyzer Software. Appropriate amounts of 20β-S were added to the sperm suspension and incubated for 10 min before sperm motility was measured. 20β-S was purchased from Steraloids Inc. Statistical significance was evaluated by two-tailed Student's *t*-test.

**Flow cytometry.** Testes were isolated from the adult *zebrafish* and dissociated by 0.2% collagenase. The cell suspension was fixated in 2% PFA and was stained with propidium iodide (PI) staining solution (50 μg/ml PI and 20 μg/ml RNase) for at least 10 min at room temperature in the dark. The cells were subsequently filtered through a 40-μm nylon mesh, and the suspension was analyzed using a FACSAria Fusion flow cytometer (BD Biosciences). Statistical significance was evaluated by two-tailed Student's *t*-test.

**Measurement of fertilization rate and survival rates in early embryogenesis.** Embryos that developed past the epiboly stage were measured as fertilized eggs. Daily survival rates were measured relative to the number of eggs reaching the epiboly stage. Statistical significance was evaluated by two-tailed Student's *t*-test.

**Reporting summary.** Further information on research design is available in the Nature Portfolio Reporting Summary linked to this article.

## Data availability

The video data are available from the authors upon request. The source data behind the figures can be found in Supplementary Data 1.

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

## Acknowledgements

We are grateful to Mrs. Natsuko Okuda at OMPU for her excellent care of *zebrafish* and *medaka* used in this study. We are grateful to medical students at OMPU, Mr. Akihiro Tani, Ms. Yuri Ozaki, Mr. Tomohiro Hirayama, Mr. Takeyoshi Murakami, Mr. Kei Yotsumoto and Mr. Kakeru Terada in particular, for their assistance in some of the experiments. We thank Dr. Toshiya Nishimura for critically reading the manuscript. This research was partially supported by Grant-in-Aid for Scientific Research (19K06460, Y.Y.).

## Author contributions

Y.Y. carried out the experiment and analyzed the data. D.H. carried out the LC–MS/MS analysis. Y.Y. designed the study. F.O. supervised the study. Y.Y. and F.O. wrote the paper.

## Competing interests

The authors declare no competing interests.
