## [Peer Review File · Communications Biology]

Trpv4-mediated apoptosis of Leydig cells induced by high temperature regulates sperm development and motility in zebrafishReviewers' comments:

Reviewer #1 (Remarks to the Author):

In this paper, the authors investigated spermatogenesis at high temperature in zebrafish and found that apoptosis of Leydig cells precedes apoptosis of germ cells at 34°C. Then, they examined the expression patterns of several trp genes in the testis and found that *trpv4*, *trpm4b2*, and *trpm4b3* were upregulated, with *trpv4* being most prominently upregulated. Therefore, the authors made a knockout of *trpv4* and examined the effects of high temperature. The knockout did not decrease the number of Leydig cells, nor did its markers, *star*, *insl3*, and *cyp11c1*. When sperm motility was examined, they found that sperm motility decreased at high temperature, but not in the mutant.

In order to correlate the Leydig cell function on sperm motility, they examined the production of steroid hormone (17,20 β ,21-trihydroxy-4-pregnen-3-one, 20 β -S), which are thought to control sperm motility, and the effect on sperm motility after exposure. However, the authors only showed that 20 β -hsd decreased by high-temperature treatment in the wild type, but not in the mutant, since the amount of steroid production was below the limit of detection. Figure 3 shows decrease of the expression of *cyp17a1* and *hsd-3b* in the mutant. Therefore, it is possible that 20 β -S is also decreased in the mutant. Furthermore, the exposure experiment is very short as about 20 minutes (the material and method says 10 minutes). It is difficult to compare the effects of the 1 day in this short period. The evidence that mutant spermatozoa have restored fertilization and normal development is required.

Minor points

L118. endoplasmic reticulum (ER).

L172. 17,20 β ,21-trihydroxy-4-pregnen-3-one. Hydroxyl group has α -coordination and β -coordination. Please mention it.

L260-266. I don't quite understand the discussion here. Could you please revise it?

Figure 3B. The difference between *star*, *insl3*, *cyp11c1* and *cyp17a1* and *hsd-3b* may reflect the heterogeneity of Leydig cells. Please discuss it.

Figure 5F. Please indicate the temperature after fertilization in this experiment. High temperature seems to affect the chromosomal structure of the sperm head. If authors can show the segregation pattern of chromosomes after fertilization, I think it will be good evidence.

Reviewer #2 (Remarks to the Author):

The paper by Yamamoto and co-authors describes a mechanism to prevent fertilization in high temperature environments by Leydig cells. This mechanism, exclusive to zebrafish, is driven by *Trpv4*. In the present work, we describe through a robust experimental design the involvement of *trpv4* in the initiation of Leydig cell apoptosis, resulting in the arrest of spermatogenesis at a spermatocyte stage.

However, there are several issues that should be addressed to improve the current version of the manuscript.

Line 60 and Introduction: *Trpv4* is not properly introduced, so the thread in the introduction of why they focus on this non-selective cation channel is not understood.

Pioneering studies in zebrafish, such as Amato et al, 2012, where the presence of this protein in the sensitive organs of the fish was investigated. In addition, it would be important to discuss the

phenotype of the *trpv4* mutant, as it could potentially lose the mechanisms to correctly detect environmental changes.

The studies by Lee et al. (2018) in chum salmon and Hori et al. (2022) in oocytes are also not presented or discussed.

Reference:

- Amato, V., Vina, E., Calavia, M. G., Guerrero, M. C., Laurà, R., Navarro, M., ... & Vega, J. A. (2012). TRPV4 in the sensory organs of adult zebrafish. *Microscopy research and technique*, 75(1), 89-96.

- Lee, H. J., Lee, S. Y., & Kim, Y. K. (2021). Molecular characterization of transient receptor potential vanilloid 4 (TRPV4) gene transcript variant mRNA of chum salmon *Oncorhynchus keta* in response to salinity or temperature changes. *Gene*, 795, 145779.

Hori, S., Sakamoto, N., & Saitoh, O. (2022). Cloning and functional characterization of medaka TRPV4. *Comparative Biochemistry and Physiology Part A: Molecular & Integrative Physiology*, 267, 111182.

Line 86: This sentence should be rewritten, since the decrease in Leydig cells is then specifically observed in Figure 1B-C.

Line 86 and Figure 1A: It is important to add the photos of the testes kept at 29°C at the different times studied, to determine that there were no morphological changes in the control during the treatment period. This could be added as a supplementary figure.

Line 254: Similar results were recently found in medaka (Moreno Acosta, et al., 2023), where exposure to high temperature slowed the passage of spermatocytes to spermatids through activation of the Notch pathway, with an increase in *pen-2* expression in Leydig cells. These findings should be discussed.

Reference:

Moreno Acosta, O. D., Boan, A. F., Hattori, R. S., & Fernandino, J. I. (2023). Notch pathway is required for protection against heat stress in spermatogonial stem cells in medaka. *Fish physiology and biochemistry*, 49(3), 487–500. <https://doi.org/10.1007/s10695-023-01200-w>

Minnor:

Line 29: Replace the term "cold-blooded" by "poikilotherms " animals

Thank you for inviting us to submit a revised draft of our manuscript entitled, “Trpv4-mediated apoptosis of Leydig cells induced by high temperature regulates sperm development and motility in zebrafish” to Communications Biology. We also appreciate the time and effort dedicated to providing insightful feedback on how to strengthen our paper. We have incorporated changes that reflect the detailed suggestions reviewers have graciously provided.

Following are point-by-point responses to the questions and comments. We hope we have addressed the concerns raised by reviewers sufficiently and you will find the paper suitable for publication.

Sincerely,

Fumihito Ono, MD, PhD

Professor

Department of Physiology

Osaka Medical and Pharmaceutical University

2-7, Daigaku-machi, Takatsuki, Osaka 569-8686, JAPAN

Tel: +81-(0) 72-683-1221

E-mail: fumihito.ono@ompu.ac.jp

Yasuhiro Yamamoto, PhD

assistant professor

Department of Physiology

Osaka Medical and Pharmaceutical University

2-7, Daigaku-machi, Takatsuki, Osaka 569-8686, JAPAN

Tel: +81-(0) 72-683-1221

E-mail: yasuhiro.yamamoto@ompu.ac.jp

Reviewer #1:

In this paper, the authors investigated spermatogenesis at high temperature in zebrafish and found that apoptosis of Leydig cells precedes apoptosis of germ cells at 34°C. Then, they examined the expression patterns of several trp genes in the testis and found that trpv4, trpm4b2, and trpm4b3 were upregulated, with trpv4 being most prominently upregulated. Therefore, the authors made a knockout of trpv4 and examined the effects of high temperature. The knockout did not decrease the number of Leydig cells, nor did its markers, star, insl3, and cyp11c1. When sperm motility was examined, they found that sperm motility decreased at high temperature, but not in the mutant.

In order to correlate the Leydig cell function on sperm motility, they examined the production of steroid hormone (17,20 β ,21-trihydroxy-4-pregnen-3-one, 20 β -S), which are thought to control sperm motility, and the effect on sperm motility after exposure. However, the authors only showed that 20 β -hsd decreased by high-temperature treatment in the wild type, but not in the mutant, since the amount of steroid production was below the limit of detection. Figure 3 shows decrease of the expression of cyp17a1 and hsd-3b in the mutant. Therefore, it is possible that 20 β -S is also decreased in the mutant. Furthermore, the exposure experiment is very short as about 20 minutes (the material and method says 10 minutes). It is difficult to compare the effects of the 1 day in this short period. The evidence that mutant spermatozoa have restored fertilization and normal development is required.

We acknowledge this is a critical point. In revision, we newly analyzed the effect of high temperature treatment on fertilization rate and early development in Trpv4^{-/-} spermatozoa (Fig.5). In Trpv4^{-/-}, fertilization rate was not decreased, while developmental abnormalities were significantly increased (Figure 5 C-E). The results suggested that Trpv4 is involved in sperm motility but not in sperm quality under high temperature conditions. Related descriptions have been provided in the revised manuscript (Lines 208-215).

Minor points

L118. endoplasmic reticulum (ER).

As suggested, we have rephrased this term to “endoplasmic reticulum (ER)” (Lines 120).

L172. 17,20 β ,21-trihydroxy-4-pregnen-3-one. Hydroxyl group has α -coordination and β -coordination. Please mention it.

As suggested, we have rephrased this term to “17 α ,20 β ,21-trihydroxy-4-pregnen-3-one” (Lines 177).

L260-266. I don't quite understand the discussion here. Could you please revise it?

We apologize for the confusion caused by the discussion which may have been misleading. We rearranged the sentences to convey our ideas in the revised manuscript (Lines 273-279).

Figure 3B. The difference between *star*, *insl3*, *cyp11c1* and *cyp17a1* and *hsd-3b* may reflect the heterogeneity of Leydig cells. Please discuss it.

This is an excellent suggestion. We agree that high temperature stimulation may affect hormone secretion in *Trpv4* *-/-* and that Leydig cells in the testes of adult zebrafish may be a heterogeneous population, in view of the heterogeneity of Leydig cell reported in adult mice. In revision, we described these possibilities in the text (lines 159-160).

Figure 5F. Please indicate the temperature after fertilization in this experiment. High temperature seems to affect the chromosomal structure of the sperm head. If authors can show the segregation pattern of chromosomes after fertilization, I think it will be good evidence.

We are sorry we did not mention the temperature after fertilization in Figure 5F. This information is now added (Lines 288-289). Since the cell analyzer analysis suggested that the percentage of abnormal sperm is small, we expect that it will be difficult to detect abnormalities in chromosome structure caused by high temperature treatment.

Reviewer 2

The paper by Yamamoto and co-authors describes a mechanism to prevent fertilization in high temperature environments by Leydig cells. This mechanism, exclusive to zebrafish, is driven by Trpv4. In the present work, we describe through a robust experimental design the involvement of trpv4 in the initiation of Leydig cell apoptosis, resulting in the arrest of spermatogenesis at a spermatocyte stage. However, there are several issues that should be addressed to improve the current version of the manuscript.

Line 60 and Introduction: Trpv4 is not properly introduced, so the thread in the introduction of why they focus on this non-selective cation channel is not understood. Pioneering studies in zebrafish, such as Amato et al, 2012, where the presence of this protein in the sensitive organs of the fish was investigated. In addition, it would be important to discuss the phenotype of the trpv4 mutant, as it could potentially lose the mechanisms to correctly detect environmental changes. The studies by Lee et al. (2018) in chum salmon and Hori et al. (2022) in oocytes are also not presented or discussed.

We appreciate the constructive suggestion. We rewrote the relevant sections of the results to address the importance of Trp in zebrafish temperature sensitivity and possible contribution of sensory organs in the phenotypes of the Trpv4 knockout. Two papers suggested by the reviewer are newly cited in the revised manuscript (Lines 131-133).

Line 86: This sentence should be rewritten, since the decrease in Leydig cells is then specifically observed in Figure 1B-C.

The sentence was rewritten (Lines 88-89).

Line 86 and Figure 1A: It is important to add the photos of the testes kept at 29°C at the different times studied, to determine that there were no morphological changes in the control during the treatment period. This could be added as a supplementary figure.

We appreciate this constructive suggestion. There were no abnormalities in spermatogenesis after two weeks of incubation in the control temperature. Related descriptions and photos (Supplementary Figure 2.) have been provided in the revised manuscript (Lines 69-71).

Line 254: Similar results were recently found in medaka (Moreno Acosta, et al., 2023), where exposure to high temperature slowed the passage of spermatocytes to spermatids through activation of the Notch

pathway, with an increase in pen-2 expression in Leydig cells. These findings should be discussed.

Thank you for the helpful comment. We now cited this paper to indicate the important point that high temperature stimulation in different fish species leads to similar spermatogenesis abnormalities (Lines 230-231 and 240-241).

Minor:

Line 29: Replace the term "cold-blooded" by "poikilotherms " animals

As suggested, we rephrased this term to “poikilotherms” (Line 29).

Reviewers' comments:

Reviewer #1 (Remarks to the Author):

In response to my comments, the authors have added data on the fertilization rate of *trpv4*^{-/-} sperm and the developmental abnormalities of the fertilized individuals. However, although it was mentioned that there was no statistically significant difference in fertilization ratios between 29° *trpv4*^{+/+} and 34° *trpv4*^{-/-}, there was no mention about statistically significance difference between 34° *trpv4*^{+/+} and 34° *trpv4*^{-/-} (Fig. 5C). Therefore, it cannot be said that the fertilization rate is restored when Leydig cells do not disappear at 34°. Since the authors did not show a decrease in 20 β -S at 34°, and the 20 β -hsd expression of 34° 1day is not significantly different from those at 29° (Fig. 4G), I think this result to be an important demonstration of the involvement of Leydig cells in sperm motility at that temperature.

In the decrease of E3 spermatids at 34°, there is no difference between *trpv4*^{-/-} and *trpv4*^{+/+} (Fig. 3A). Although results have shown that recovery is earlier, it is unclear how Leydig cells contribute. Considering these points together, there is still little experimental evidence for the function of Leydig cells in spermatogenesis and sperm motility under high temperatures in this manuscript.

Reviewer #2 (Remarks to the Author):

The authors have followed the suggestions of both reviewers, which has significantly improved the present version of the manuscript.

Thank you for allowing us to submit a revised draft of our manuscript entitled, “Trpv4-mediated apoptosis of Leydig cells induced by high temperature regulates sperm development and motility in zebrafish” to Communications Biology. We also appreciate the effort reviewers dedicated to providing insightful and constructive feedback. Following are point-by-point responses to the questions and comments. We feel the new addition strengthens the paper significantly, and we hope you will find the paper suitable for publication.

Sincerely,

Fumihito Ono, MD, PhD

Professor

Department of Physiology

Osaka Medical and Pharmaceutical University

2-7, Daigaku-machi, Takatsuki, Osaka 569-8686, JAPAN

Tel: +81-(0) 72-683-1221

E-mail: fumihito.ono@ompu.ac.jp

Yasuhiro Yamamoto, PhD

assistant professor

Department of Physiology

Osaka Medical and Pharmaceutical University

2-7, Daigaku-machi, Takatsuki, Osaka 569-8686, JAPAN

Tel: +81-(0) 72-683-1221

E-mail: yasuhiro.yamamoto@ompu.ac.jp

Reviewer #1:

In response to my comments, the authors have added data on the fertilization rate of *trpv4*^{-/-} sperm and the developmental abnormalities of the fertilized individuals. However, although it was mentioned that there was no statistically significant difference in fertilization ratios between 29° *trpv4*^{+/+} and 34° *trpv4*^{-/-}, there was no mention about statistically significance difference between 34° *trpv4*^{+/+} and 34° *trpv4*^{-/-} (Fig. 5C). Therefore, it cannot be said that the fertilization rate is restored when Leydig cells do not disappear at 34°. Since the authors did not show a decrease in 20 β -S at 34°, and the 20 β -hsd expression of 34° 1day is not significantly different from those at 29° (Fig. 4G), I think this result to be an important demonstration of the involvement of Leydig cells in sperm motility at that temperature.

In the decrease of E3 spermatids at 34°, there is no difference between *trpv4*^{-/-} and *trpv4*^{+/+} (Fig. 3A). Although results have shown that recovery is earlier, it is unclear how Leydig cells contribute. Considering these points together, there is still little experimental evidence for the function of Leydig cells in spermatogenesis and sperm motility under high temperatures in this manuscript.

We apologize that we did not mention the statistically significant difference between 34° *trpv4*^{+/+} and 34° *trpv4*^{-/-} (Fig. 5C). This information is now added (Lines 208-210). These results are consistent with our proposal that zebrafish actively reduces sperm motility in order to suppress fertilization in high temperatures, which is discussed further in the newly added Supplementary Figure 12.

The reviewer's comment made us realize that we did not get across major points of our conclusion sufficiently. To overcome this shortcoming, we newly added the supplementary Figure 12, that would explain the significance of results for each experiment in the overall framework. Specifically, by separating the sperm quality and the sperm motility, we tried to show more clearly that the 20 β -S production by Leydig cells is required for the latter, but not the former.

Dear Editor,

Thank you for accepting our revised manuscript entitled "Trpv4-mediated apoptosis of Leydig cells induced by high temperature regulates sperm development and motility in zebrafish" for publication in Communications Biology.

In this revision, we added dot plot data of all graph. All items in the editorial request table were checked and changes were made where necessary.

We believe this paper will be of interest to broad readership for publication in Communications Biology. We thank you in advance for considering our manuscript, and look forward to hearing from you.

Sincerely,

Fumihito Ono, MD, PhD

Professor
Department of Physiology
Osaka Medical and Pharmaceutical University
2-7, Daigaku-machi, Takatsuki, Osaka 569-8686, JAPAN
Tel: +81-(0) 72-683-1221
E-mail: fumihito.ono@ompu.ac.jp

Yasuhiro Yamamoto, PhD

assistant professor
Department of Physiology
Osaka Medical and Pharmaceutical University
2-7, Daigaku-machi, Takatsuki, Osaka 569-8686, JAPAN
Tel: +81-(0) 72-683-1221
E-mail: yasuhiro.yamamoto@ompu.ac.jp